# Chronic Carbon Tetrachloride Applications Induced Hepatocyte Apoptosis in Lipocalin 2 Null Mice through Endoplasmic Reticulum Stress and Unfolded Protein Response

**DOI:** 10.3390/ijms21155230

**Published:** 2020-07-23

**Authors:** Erawan Borkham-Kamphorst, Ute Haas, Eddy Van de Leur, Anothai Trevanich, Ralf Weiskirchen

**Affiliations:** 1Institute of Molecular Pathobiochemistry, Experimental Gene Therapy and Clinical Chemistry, RWTH Aachen University Hospital, D-52074 Aachen, Germany; uh57@arcor.de (U.H.); eddy.vandeleur@t-online.de (E.V.d.L.); 2Department of Statistics, Faculty of Science, Khon Kaen University, Khon Kaen 40002, Thailand; anothai@kku.ac.th

**Keywords:** ER stress, UPR, LCN2, CHOP, hepatocyte, apoptosis

## Abstract

The lack of Lipocalin (LCN2) provokes overwhelming endoplasmic reticulum (ER) stress responses in vitro and in acute toxic liver injury models, resulting in hepatocyte apoptosis. LCN2 is an acute phase protein produced in hepatocytes in response to acute liver injuries. In line with these findings we investigated ER stress responses of *Lcn2*^−/−^ mice in chronic ER stress using a long-term repetitive carbon tetrachloride (CCl_4_) injection model. We found chronic CCl_4_ application to enhance ER stress and unfolded protein responses (UPR), including phosphorylation of eukaryotic initiation factor 2α (eIF2α), increased expression of binding immunoglobulin protein (BiP) and glucose-regulated protein 94 (GRP94). IRE1α/TRAF2/JNK signaling enhanced mitochondrial apoptotic pathways, and showed slightly higher in *Lcn2*^−/−^ mice compared to the wild type counterparts, leading to increased hepatocyte apoptosis well evidenced by terminal deoxynucleotidyl transferase dUTP nick end labeling (TUNEL) staining. Hepatocyte injuries were confirmed by significant high serum alanine transaminase (ALT) levels in CCl_4_-treated *Lcn2*^−/−^ mice. *Lcn2*^−/−^ mice furthermore developed mild hepatic steatosis, supporting our finding that ER stress promotes lipogenesis. In a previous report we demonstrated that the pharmacological agent tunicamycin (TM) induced ER stress through altered protein glycosylation and induced high amounts of C/EBP-homologous protein (CHOP), resulting in hepatocyte apoptosis. We compared TM-induced ER stress in wild type, *Lcn2*^−/−^, and *Chop* null (*Chop*^−/−^) primary hepatocytes and found *Chop*^−/−^ hepatocytes to attenuate ER stress responses and resist ER stress-induced hepatocyte apoptosis through canonical eIF2α/GADD34 signaling, inhibiting protein synthesis. Unexpectedly, in later stages of TM incubation, *Chop*^−/−^ hepatocytes resumed activation of IRE1α/JNK/c-Jun and p38/ATF2 signaling, leading to late hepatocyte apoptosis. This interesting observation indicates *Chop*^−/−^ mice to be unable to absolutely prevent all types of liver injury, while LCN2 protects the hepatocytes by maintaining homeostasis under ER stress conditions.

## 1. Introduction

Lipocalin-2 (LCN2), also known as neutrophil gelatinase-associated lipocalin (NGAL), is a 24 kDa secretory glycoprotein isolated from human neutrophils [1]. LCN2 is an acute-phase protein in response to harmful conditions such as endotoxin-inflammatory stimuli, oxidative stress, and acute endoplasmic reticulum (ER) stress [2,3,4]. Furthermore, LCN2 plays a well-documented critical role in host defenses against bacterial infections by sequestering iron-containing siderophores [5,6]. In experimental animal models, LCN2 protects against hepatotoxins-induced acute liver injury and promotes liver regeneration from partial hepatectomy [7,8]. However, LCN2 does also promote liver injury and inflammation in alcoholic steatohepatitis (ASH) and nonalcoholic steatohepatitis (NASH) [9,10,11]. Both hepatocytes and neutrophils are important cell types that produce LCN2 [1,7,8,12]. We have previously shown that hepatocytes are the major cell type for LCN2 and that LCN2 protects hepatocytes from IL-1β-induced stress [8,13]. LCN2 from both neutrophils and local epithelial cells is required for optimal resistance against the dissemination of a siderophore-producing pathogen infection [14]. Recently, we reported that hepatocytes from *Lcn2* null mice developed an overwhelming ER stress response in vitro and in acute toxic liver injury models [4].

ER stress is a physiological process activating a set of signaling pathways, called unfolded protein response (UPR). In this process, three specific ER-localized protein sensors are relevant in the execution of respective responses. These are the inositol-requiring enzyme 1α (IRE1α), the double-stranded RNA-dependent protein kinase (PKR)-like ER kinase (PERK), and the activating transcription factor 6 (ATF6) [15,16]. The downstream stimulated branches promote cell survival by reducing misfolded protein levels. However, if this machinery is overloaded, the signaling turns to induce apoptotic cell death [17,18,19,20].

Repeated administration of carbon tetrachloride (CCl_4_) is one of the most commonly used experimental models for inducing toxin-mediated liver fibrosis. The toxin is metabolized in the liver by the cytochrome P450 system, which forms the trichloromethyl radical (CCl_3_*). This aggressive radical chemically attacks nucleic acid, protein, and lipids. It further interferes with lipid metabolism and homeostasis and provokes hepatic steatosis. DNA adducts further provoke somatic mutations, ending in hepatocellular carcinoma (HCC). CCl_3_* can further react with oxygen, forming the trichloromethylperoxy radical CCl_3_OO* that can initiate significant lipid peroxidation and destruction of polyunsaturated fatty acids [21,22]. This results in increased mitochondrial permeability, reduced integrity of ER and plasma membranes, dysfunctions of cellular calcium homeostasis, and cell damage. Therefore, this model is suitable for further evaluation of *Lcn2* null (*Lcn2*^−/−^) mice in chronic ER stress response. We found that although the markers of ER stress were not different between the wild type and *Lcn2*^−/−^ mice, *Lcn2*^−/−^ mice showed more hepatocyte apoptosis compared to the wild type counterpart. In our previous report, we demonstrated that the pharmacological agent TM induced ER stress through altered protein glycosylation. TM treatment resulted in large quantities of de-glycosylation LCN2 in wild type hepatocytes and induced huge amounts of CHOP protein in *Lcn2*^−/−^ mice, resulting in hepatocyte apoptosis [4]. We therefore decided to extend our studies and investigate whether *Chop* null (*Chop*^−/−^) hepatocytes were able to prevent TM-induced apoptosis. We found *Chop*^−/−^ hepatocytes to attenuate and delay ER stress responses, especially in the IRE1α signaling pathway, resulting in attenuating hepatocyte apoptosis. However, in later stages of TM incubation, *Chop*^−/−^ hepatocytes resumed activation of IRE1α/JNK/c-Jun and p38/ATF2 signaling, leading to late hepatocyte apoptosis.

## 2. Results

### 2.1. Repeated CCl_4_ Intoxication Induced ER Stress and UPR

In response to repeated CCl_4_ administration, both wild type and *Lcn2*^−/−^ mice exert ER stress and UPR, as evidenced by the occurrence of spliced X-box-binding protein 1 (*Xbp1s*) mRNA 48 h upon CCl_4_ administration (Figure 1A). The hepatic mRNA levels of several ER stress markers (*Grp94*, *Atf4*) were significantly higher in *Lcn2*^−/−^ mice in relation to oil- or CCl_4_-injected wild type mice (Figure 1B). However, subject ER stress markers were comparable at the protein level between wild type and *Lcn2*^−/−^ mice with increased GRP94, and p-eIF2α protein expression in CCl_4_-treated groups. Moreover, we found a tendency for elevated p21 in CCl_4_-treated groups, while CHOP levels fluctuated between animals (Figure 1C,D).

### 2.2. Chronic Repeated CCl_4_ Administration Activated c-Jun N-Terminal Kinase (JNK) and Intrinsic Hepatocyte Apoptosis

In response to ER stress, IRE1α initiates a signaling pathway that activates JNK via the cytoplasmic part of IRE1α bound TRAF2, an adaptor protein that couples the plasma membrane receptors [23]. We found significant JNK phosphorylation in chronic CCl_4_-treated mice that was slightly higher in *Lcn2*^−/−^ mice compared to the wild type, while TRAF2 activation was similar (Figure 2A,B). JNK further regulated gene expression through the phosphorylation and activation of downstream transcription factor c-Jun (Figure 2A,B).

ER, plasma membrane, mitochondria, and Golgi apparatus are the main subcellular structures of hepatocytes affected by CCl_4_ exposure. In CCl_4_-treated *Lcn2*^−/−^ mice we found a significant upregulation of mitochondrial protein Bax and cytochrome c, with slightly decreased Bcl2, while showing a compensated increase of Bcl-xL (Figure 2C,D). Caspase-9 activation was significantly higher in both oil- and CCl_4_-treated *Lcn2*^−/−^ mice, while cleaved caspase-3 levels and TRB3 were similar in the CCl_4_-treated groups. No significant differences were observed for p53-upregulated modulator of apoptosis (PUMA) (Figure 2C,D). Immunohistochemistry staining for cleaved caspase-3 found positive cells distributed among non-parenchymal cells from mineral oil administration. Livers of CCl_4_-treated wild type animals showed positive in infiltrating macrophages and inflammatory cells around the hepatic central veins, while in *Lcn2^−/−^* animals cleaved caspase-3 was found in surrounding hepatocytes (Figure 2E). We further confirmed apoptosis using terminal deoxynucleotidyl transferase dUTP nick end labeling (TUNEL) assay and found that the apoptotic cell types were different between wild type and *Lcn2* null animals. In wild type livers, TUNEL-positive cells were infiltrating inflammatory cells located around hepatic central veins, while in *Lcn2* null livers TUNEL-positive cells were mainly found scattered in hepatocytes zone 3 (Figure 2F). These findings indicate that the *Lcn2* null hepatocytes are sensitive to chronic CCl_4_-induced intoxication, in part through ER stress-induced apoptosis. Subject hepatocyte damage was confirmed by significantly increased serum aspartate aminotransferase (AST) and alanine aminotransferase (ALT) in *Lcn2* null mice, while serum albumin levels were lower in both groups of CCl_4_-treated mice (Figure 2G).

### 2.3. Lcn2 Null Mice Developed Slightly More Steatosis Than Wild Type Mice

CCl_4_ administration increases microsomal lipids as early as three hours due to reduced secretion of triglycerides from the ER into plasma [22]. We found Oil Red O staining to show more fat droplets in liver sections of *Lcn2* null mice compared to the wild type (Figure 3A). Serum triglyceride and cholesterol levels were higher in CCl_4_-treated mice. Serum triglyceride levels in CCl_4_-treated *Lcn2*^−/−^ mice were slightly lower than in wild type mice due to triglyceride accumulation and steatosis. No changes were observed in serum high-density lipoproteins (HDL) and low-density lipoproteins (LDL) (Figure 3B). Foregoing is in line with previous reports, showing increased lipid droplets in a single dose CCl_4_ injection and in hepatocytes treated with TM and TG [4,24].

*Lcn2*^−/−^ hepatocytes treated with TM showed huge amounts of the CHOP protein, which plays an important role in ER stress-induced apoptosis. In consequence, we employed cultured primary hepatocytes from *Chop*^−/−^ mice subjected to TM or TG to verify whether *Chop*^−/−^ would be able to protect hepatocyte apoptosis as compared to wild type and *Lcn2*^−/−^ hepatocytes. We found *Lcn2*^−/−^ hepatocytes to detach from the culture plates and less confluent than wild type. By contrast, *Chop*^−/−^ hepatocytes clearly showed minimal hepatocyte apoptosis (Figure 3C). Oil Red O staining in *Lcn2*^−/−^ showed significant high amounts of lipid droplets, with more modest levels in *Chop* null hepatocytes (Figure 3D).

### 2.4. Strongest ER Stress Responses Were Observed in Lcn2^−/−^ Hepatocytes That Also Expressed High Levels of Cyclic AMP-Responsive Element-Binding Protein 3-Like 3 (CREB3L3)

We further comparatively evaluated the responses to TM- and TG-induced ER stress and UPR in cultured hepatocytes isolated from wild type, *Lcn2*^−/−^, and *Chop*^−/−^ mice and compared it to those obtained after IL-1β stimulation. TM induced significantly higher levels of ER stress marker genes Grp94, Bip, and Chop in *Lcn2*^−/−^ hepatocytes, together with high amounts of *Xbp1s* mRNA, as compared to wild type and *Chop*^−/−^ hepatocytes (Figure 4A,B). The ER stress marker proteins GRP94, BiP, and IRE1α showed highest expression in *Lcn2*^−/−^ and lowest expression in *Chop*^−/−^ hepatocytes, along with slightly decreased full length ATF6 in all types of hepatocytes. Surprisingly, we observed stronger de-phosphorylation of eIF2α in wild type and *Lcn2*^−/−^ than in *Chop*^−/−^ hepatocytes, while the ATF4 transcription factor remained at higher levels compared to *Chop*^−/−^ hepatocytes. The baseline levels of phosphorylated and total eIF2α were higher in *Chop*^−/−^ hepatocytes and lower in GADD34 compared to wild type and *Lcn2*^−/−^, pointing at the compensatory feedback from the lack of CHOP protein. TM induced huge amounts of CHOP protein in *Lcn2*^−/−^ hepatocytes and no CHOP protein in *Chop*^−/−^ hepatocytes, with lower p21 in all types (Figure 4C).

CREB3L3 is a hepatocyte-specific transcription factor (CREBH) localized at the ER membrane and activated through S1P and S2P proteolytic cleavage during ER stress. CREBH is a key metabolic regulator of hepatic lipogenesis, fatty acid oxidation, and lipolysis under metabolic stress. CREBH also promotes lipid droplet enlargement and triglyceride accumulation in liver [25,26]. We found *Lcn2*^−/−^ hepatocytes to show significant higher CREBH in both mRNA and protein levels, while TM incubation did activate proteolytic cleavage of full-length CREBH protein (Figure 4D).

### 2.5. Pharmacological ER Stress Inducer Tunicamycin Activates TRAF2, c-Jun N-Terminal Kinase, NF-κB, and Mitochondrial Apoptotic Proteins in Hepatocytes

Following 24 h TM incubation, hepatocytes showed declined TRAF2 and JNK phosphorylation, but JNK signaling leading to c-Jun phosphorylation was unaltered. This was more pronounced in *Lcn2*^−/−^ hepatocytes, while NF-κB p65 was phosphorylated in all types (Figure 4D). Additionally, IL-1β induced the highest levels of LCN2, whereas TM treatment resulted in high quantities of de-glycosylated LCN2. Surprisingly, *Chop*^−/−^ hepatocytes significantly attenuated IL-1β and TM-induced LCN2 production (Figure 4D), suggesting that LCN2 in part is a downstream target of CHOP, and may explain the high amounts of CHOP accumulation in TM-treated *Lcn2*^−/−^ hepatocytes.

ER stress pathways through JNK activation lead to inhibition of the anti-apoptotic proteins (Bcl-2 and Bcl-xL) and activation of the pro-apoptotic proteins (Bax, Bak, Bim, Bad, and Bid), eventually leading to increased mitochondrial permeabilization required for hepatocyte apoptosis. Treatment with TM for 24 h resulted in decreased Bcl-xL levels, combined with upregulation of Bax, Bak, and Cytochrome c in *Lcn2*^−/−^ hepatocytes, compared to wild type and *Chop*^−/−^ (Figure 4E).

### 2.6. Chop^−/−^ Alternates ER Stress Responses, Protects Against Tunicamycin-Activated ER Stress, and Induces Hepatocyte Apoptosis Through Delayed JNK Activation

As we were unable to detect the apoptotic markers cleaved Caspase-9 and -3 in hepatocytes treated for 24 h with TM or TG, we prolonged incubation to 48 and 72 h. In the later stage of ER stress, activated IRE1α induced *Xbp1s* mRNA levels that remained persistent. The levels of *Xbp1s* mRNA became almost level in all type hepatocytes at 48 h, with only slightly higher values in *Lcn2*^−/−^, compared to wild type and *Chop*^−/−^ hepatocytes (Figure 5A). Expression of ER stress marker proteins BiP, GRP94, IRE1α, and PRRK was the lowest in *Chop*^−/−^ hepatocytes combined with increased levels of phosphorylated eIF2α and ATF4. TM incubation for 48 and 72 h resulted in a clear upregulation of GADD34 phosphatase that was slightly less prominent in *Chop*^−/−^ hepatocytes (Figure 5B). CHOP induction in *Lcn2*^−/−^ hepatocytes remained higher than in the wild type at 48 h, declining at 72 h with no CHOP protein in *Chop*^−/−^ hepatocytes. Accumulation of LCN2 protein was significant lower in *Chop*^−/−^ compared to wild type hepatocytes (Figure 5B).

Upon 48 and 72 h TM incubation, *Xbp1s* mRNA levels remained persistently high. Therefore, we examined the IRE1 kinase signaling via TRAF2, NF-κB, and JNK. NF-κB p65 was activated in all types of hepatocytes, with the highest values in those obtained from *Lcn2*^−/−^ animals. By contrast, however, JNK activation was unexpectedly high in *Chop*^−/−^ hepatocytes together with downstream c-Jun phosphorylation, but markedly lower in hepatocytes isolated from wild type and *Lcn2*^−/−^ mice. Additionally, p38 activation was significant higher in *Chop*^−/−^ hepatocytes, while ATF2 phosphorylation was found unaltered in all types (Figure 5C). The ATF2 transcription factor under sustained stress conditions is phosphorylated by the stress-activated MAPK JNK and p38 [27,28]. These findings indicate that the lack in CHOP delayed IRE1α signaling and resulted in delayed apoptosis in *Chop*^−/−^ hepatocytes.

Since Bcl-2 family proteins are involved in the ER stress-mediated apoptotic pathways through JNK activation, we next analyzed the expression of these proteins. TM incubation for 48 and 72 h did indeed lead to decreased Bcl2 and Bcl-xL levels in all types of hepatocytes, but increased pro-apoptotic Bax protein only in *Lcn2*^−/−^. Furthermore, levels of the pro-apoptotic BH3-only proteins BIM and PUMA showed marked upregulation in *Lcn2*^−/−^ hepatocytes compared to wild type and *Chop*^−/−^. The levels of Bim correlated well with the increase of cleaved Caspase-3, acting as the executor caspase during hepatocyte apoptosis, with the highest expression found in *Lcn2*^−/−^ hepatocytes and the lowest in *Chop*^−/−^ hepatocytes (Figure 5D).

### 2.7. ER Stress and UPR in Early Stage Of Tunicamycin-Stimulated Primary Hepatocytes

To verify *Chop*^−/−^ hepatocytes delayed JNK activation, we next looked at the early period of TM stimulation and found *Xbp1s* mRNA production to start 2 h upon TM incubation (Figure 6A). Phosphorylated eIF2α levels were detected already at 1 h and increased persistently, followed by ATF4 and CHOP at 4 and 8 h, respectively. Upregulation of BiP and GRP94 was noticed at 8 and 24 h, while the block of LCN2 glycosylation started at 1 h and progressed over time (Figure 6B). The amount of p-JNK registered highest at 1 h, with continuous decline thereafter, while downstream p-c-Jun increased at 2 h, followed by a slow decline in correspondence to the levels of p-JNK (Figure 6C).

We next comparatively analyzed effects of TM-treatment in *Chop*^−/−^ and wild type hepatocytes under conditions established before [4]. When compared *Chop*^−/−^ to wild type hepatocytes during early TM stimulation after 1, 2, and 4 h, the levels of p-JNK and p-p38 did decline over time (Figure 6D,E). Compared to the wild type, *Chop*^−/−^ hepatocytes showed significantly lower amounts of p-c-Jun, while phosphorylation of extracellular signal-regulated kinase 1/2 (ERK1/2) was higher. Activation of the ERK1/2 confers hepatocyte resistance to death, whereas sustained JNK/c-Jun/AP-1 activation promotes apoptosis. These findings may in part explain *Chop*^−/−^ hepatocytes’ resistance to TM-induced apoptosis.

## 3. Discussion

Chronic CCl_4_ administration induces ER stress in wild type and *Lcn2*^−/−^ mice. The mRNA expression of ER stress markers Grp94, Bip, ATF4, and Chop were slightly higher in *Lcn2*^−/−^ mice compared to the wild type mice, but showed no significant difference in protein levels (Figure 1). However, JNK activation and downstream transcription factor c-Jun phosphorylation in the liver was higher in *Lcn2*^−/−^ mice. We found upregulation of mitochondrial protein Bax and Cytochrome c, with slightly decreased Bcl2 in CCl_4_-treated *Lcn2*^−/−^ mice resulting in Caspase-9 activation. Moreover, cleaved Caspase-9 showed significantly higher levels in both oil- and CCl_4_-treated *Lcn2*^−/−^ mice (Figure 2), indicating that ER stress-mediated cell death is executed by the canonical mitochondrial apoptosis pathway, where the Bcl2 family plays a central role. TUNEL staining showed the apoptotic cell types to be different between wild type and *Lcn2* null animals. In wild type livers, TUNEL-positive cells were confined to infiltrating inflammatory cells around hepatic central veins, while *Lcn2* null livers showed TUNEL-positive cells scattered in hepatocytes zone 3 (Figure 2). Subject hepatocyte damage was confirmed by higher serum AST and ALT in *Lcn2* null mice, indicating that these hepatocytes are more sensitive to chronic CCl_4_ intoxication, in part through ER stress-induced apoptosis. ER stress contributes to steatosis by activating lipogenic pathways in hepatocytes through induction of sterol regulatory element-binding protein (SREBP) induction [29,30]. Long-term repeated CCl_4_ application induced mild steatosis in *Lcn2*^−/−^ livers, in line with TM- or TG-induced ER stress and steatosis in cultured *Lcn2*^−/−^ hepatocytes (Figure 3). Additionally, high levels of CREBH transcription factor in *Lcn2*^−/−^ hepatocytes can be activated by ER stress to promote lipid droplet enlargement and hepatic triglyceride accumulation [25,26].

Upon TM stimulation, *Lcn2*^−/−^ hepatocytes produced very high levels of the pro-apoptotic protein CHOP [4]. We therefore compared the ER stress responses of TM-induced cultured primary hepatocytes isolated from wild type, *Lcn2*^−/−^, and *Chop*^−/−^ mice. We found *Chop*^−/−^ hepatocytes to attenuate ER stress and UPR, as evidenced by decreased ER stress markers, mRNA, and protein levels (Figure 4). The IRE1α branch signaling especially showed minimal *Xbp1s* mRNA at 24 h with decreased IRE1α protein and JNK/c-Jun activation. Hepatocytes lacking CHOP showed decreased GADD34, a specific eIF2α phosphatase, resulting in higher baseline levels of p-eIF2α and total eIF2α in *Chop*^−/−^ compared to wild type and *Lcn2*^−/−^ hepatocytes. Therefore, the protein production in *Chop*^−/−^ hepatocytes remains inhibited in order to minimize ER overloading. This seems to explain why cultured *Chop*^−/−^ hepatocytes showed minimal apoptosis (Figure 3C) compared to *Lcn2*^−/−^ that contained high amounts of CHOP, pointing to a CHOP-mediated interplay between protein synthesis and cell death during ER stress [4,31]. Interestingly, upon 48 h TM incubation, the levels of *Xbp1s* mRNA quantities were almost identical in all types of hepatocytes (Figure 5A) and correlated with increased JNK/c-Jun activation in *Chop*^−/−^ hepatocytes at 48 h with a slight decline at 72 h, while JNK/c-Jun signaling in wild type and *Lcn2*^−/−^ was already minimal. Delayed IRE1α-mediated JNK activation in *Chop*^−/−^ hepatocytes represents a mechanism by which IRE1α can manipulate between pro- and anti-apoptotic signals, tipping the balance in favor of apoptosis. The later state of JNK/c-Jun activation may explain previous findings showing that *Chop*^−/−^ primary mouse embryo fibroblasts still undergo apoptosis in response to prolonged ER stress, albeit with much lower kinetics [31], and *Chop* null mice failed to protect mice from acute CCl_4_-induced liver injury [32].

ER stress-induced cell death via ATF4/CHOP or IRE1/JNK is widely accepted to occur through the mitochondrial apoptotic pathway [33,34]. We found *Lcn2*^−/−^ hepatocytes with high amounts of CHOP showed decreased Bcl-xL, the pro-survival proteins of the Bcl2 family, with increased Bax and Bak, the pro-apoptotic proteins that oligomerize and form mitochondrial pores to promote Cytochrome c release, followed by caspase-9 and -3 activation resulting in apoptosis. Whereas anti-apoptotic Bcl-2 family proteins such as Bcl-2 and Bcl-xL inhibit pore formation, a group of pro-apoptotic BH3-only proteins activate Bax and Bak directly or indirectly to induce formation of the mitochondrial pores and the release of Cytochrome c. Sustained activation of UPR signals can result in upregulation of pro-apoptotic Bcl-2 family members such as BIM and PUMA. Upon 48 and 72 h TM incubation, we detected up-regulation of BIM and PUMA, showing the highest values in *Lcn2*^−/−^ hepatocytes. BIM and PUMA are induced through ATF4/CHOP and CHOP/AP-1, respectively [35,36]. Both BH3-only protein levels correlated well with the highest CHOP levels in wild type and *Lcn2*^−/−^ hepatocytes.

In sum, *Lcn2* null mice hepatocytes provide stronger TM-induced ER stress responses than hepatocytes isolated from wild type animals, particularly apparent in higher induction of CHOP (Figure 4C). The response to ER stress is further more prominent in the in vitro situation than in our in vivo experiments. We have previously shown that LCN2 acts as a protective factor in liver homeostasis [7], explaining why cells lacking LCN2 are more prone to injury and more susceptible to ER stress-stimulating triggers such as TM. However, in vivo several counteracting mechanisms might partially compensate for the harmful consequences induced by loss of LCN2, thereby preventing overshooting experimental damage.

Furthermore, LCN2 seems to have many and partially contrary biological functions. Previous work from others has shown that LCN2 exacerbates steatohepatitis in two animal models of non-alcoholic steatohepatitis by promoting neutrophil-macrophage crosstalk [11]. On the contrary, hepatocyte-derived LCN2 protects against diet-induced nonalcoholic fatty liver disease by regulating lipolysis, fatty acid oxidation, de novo lipogenesis, and apoptosis [37] or by interfering with expression of the lipid droplet associate protein Perlipin 5 (PLIN5) [38]. Moreover, LCN2 plays a protective role against the progression of hepatocellular carcinoma by suppressing cell proliferation and invasion [39], suggesting that the functionality of LCN2 depends on the condition analyzed. Nevertheless, it should be kept in mind that LCN2 was originally isolated as an acute phase protein [2], which are important mediators produced by the liver during acute and chronic inflammatory states. In general, these factors are regarded as protective factors with functions in the opsonization of micro-organisms and their products, activating the complement system, binding cellular remnants, neutralizing enzymes, scavenging free radicals, and in modulating the host’s immune response. Therefore, LCN2 should have more protective than harmful influences on liver health and homeostasis.

## 4. Materials and Methods

### 4.1. Animal Experiments and Specimen Collections

All animal protocols complied with the guidelines for animal care approved by the German Animal Care Committee (Landesamt für Naturschutz, Umwelt und Verbraucherschutz Nordrhein-Westfalen (LANUV) located in Recklinghausen, Germany; https://www.lanuv.nrw.de). Permit numbers for respective animal protocols are: 84-02.04.2012.A092 (approved 15 August 2012) for CCl_4_ injection experiments and 84-02.04.2015.A028 (approved 4 March 2015) for hepatocyte isolation from murine livers.

In our study, we used 6–8-week-old C57BL/6 wild type and *Lcn2*^−/−^ mice to investigate CCl_4_-induced ER stress. To do so, we injected mice intraperitoneally with 0.8 mL CCl_4_/kg body weight diluted in mineral oil twice a week for eight consecutive weeks. Thereafter, the animals were euthanized. Blood was drawn by heart puncture and liver specimens snap-frozen in liquid nitrogen for later analysis. Frozen tissue sections were preserved in Tissue-Tek (Sakura Finetek Europe B. V., Alphen aan den Rijn, The Netherlands) solved in in ice-cold 2-methylbutane (Roth, Karlsruhe, Germany). The samples were stored at −80 °C, or alternatively fixed in 4% buffered paraformaldehyde for subsequent immunohistological stainings.

### 4.2. RNA Isolation, cDNA Synthesis, and Real-Time Quantitative Polymerase Chain Reaction (RT-qPCR)

Total liver tissue RNA, chloroform extraction, DNAse digestion, and RNeasy cleanup was done essentially as described before [4,7]. The online ProbeFinder Software (Universal Probe Library Assay Design Center, Roche, Mannheim, Germany) was used to design primers for RT-qPCR analysis. First-strand cDNA and PCR conditions in TaqMan analysis were done as reported previously [4,7] using primers listed in Table 1. For conventional polymerase chain reaction (PCR), the cycling conditions were as follows: 30 s denaturation at 95 °C, 30 s annealing at 57 °C (*Xbp1*, 30 cycles) or 60 °C (*Gapdh*, 20 cycles), and 1 min extension at 72 °C. The amplified PCR products were separated by agarose gel electrophoresis (3%). The calculated sizes of amplification products were 170 bp for spliced *Xbp1* (*Xbp1s*) and 205 bp for unspliced *Xbp1* (*Xbp1u*), respectively.

### 4.3. SDS-PAGE and Western Blot Analysis

Liver tissues or cell lysates were prepared in RIPA buffer as previously described [3,7]. In the extracts, the protein concentrations were quantified using Bio-Rad kits according to the manufacturer’s instructions. Equal amounts of protein extracts were analyzed 4–12% Bis-Tris gradient gels, using MOPS or MES running buffers (all from Invitrogen). Subsequently, the proteins were electroblotted onto nitrocellulose membranes (Schleicher & Schuell BioScience GmbH, Dassel, Germany). After protein transfer, the membranes were stained with Ponceau S to document equal protein loading. Thereafter, non-specific binding sites on the membrane were blocked with 5% (*w*/*v*) non-fat milk powder in Tris-buffered saline and 0.1% Tween 20 (TBST). Primary antibodies (Table 2) were visualized using horseradish peroxidase conjugated secondary antibodies directed against mouse-, rabbit-, or goat IgG (Santa Cruz Biotech, Santa Cruz, CA, USA) and the SuperSignal chemiluminescent substrate (Pierce, Bonn, Germany).

### 4.4. Immunohistochemistry

Liver tissue sections were deparaffinized and rehydrated with xylene and decreasing graded ethanol, whereas antigen retrieval was engendered by heating the sections in 10 mM sodium citrate buffer (pH = 6) in a microwave for 10 min sub boiling. Slides were blocked for nonspecific binding with 3% H_2_O_2_ for 10 min, avidin-biotin (DAKO, Hamburg, Germany), followed by 5% serum of the species of which the second antibody was made in 1% bovine serum albumin (BSA), and 0.05% Tween 20 in phosphate-buffered saline (PBS). Primary antibodies were diluted in 1% BSA in PBST to concentrations of 2–5 μg/mL and incubated at 4 °C overnight, while omitting primary antibody was used for negative control. Sections were incubated with biotinylated secondary antibodies (DAKO), followed by avidin-biotin conjugated peroxidase (VECTASTAIN^®^ Elite ABC-HRP peroxidase kit (PK-6100), Vector Laboratories, Burlingame, CA, USA) and 3,3′-diaminobenzidine substrate (SIGMA*FAST*™, Sigma-Aldrich, Taufkirchen, Germany, #D4293).

### 4.5. Terminal Transferase dUTP Nick End-Labeling Assay (TUNEL)

Both paraffin-embedded and frozen sections were used for DNA fragmentation detection using the In Situ Cell Death Detection Kit, Fluorescein (Roche) and protocols we have used before [7].

### 4.6. Oil Red O Staining

For Oil Red O staining, we used air-dried and thawed frozen sections or cultured hepatocytes. Respective samples were fixed in 4% paraformaldehyde solution for 20 min and rinsed three times in deionized water to remove excess formaldehyde. Thereafter, the sections were stained in Oil Red O (Sigma-Aldrich) solution, for 30 min at room temperature. Finally, the sections were extensively washed in deionized water, counterstained in hematoxylin solution (DAKO), rinsed with running tap water for 10 min, and mounted in aqueous mounting solution (PermaFluor, Thermo Scientific, Bonn, Germany).

### 4.7. Primary Liver Cell Isolation and Culturing

Primary hepatocytes were isolated from livers of 8–12-week-old mice (*Lcn2*^−/−^, *Chop*^−/−^, wild type) bred on a C57BL/6 background using protocols described before [40]. Culturing of cells was done on collagen-coated dishes in Hepatozyme-SFM medium (Thermo Fisher Scientific, Schwerte, Germany). *Chop*^−/−^ mice with C57BL/6 background originated from Jackson Laboratory (Bar Harbor, ME, USA) [41]. The protocol used for hepatocyte isolation was approved by the German Animal Care Committee as mentioned above.

### 4.8. Tunicamycin and Thapsigargin Treatment

For these experiments, primary hepatocytes were cultured for one or two days after initial plating. The medium was changed and 0.5–1 μg/mL TM or 0.5–1 μM TG (#17765 and #T9033, Sigma-Aldrich) was added. The cells were then further incubated for additional 24, 36, 48, and 72 h. The cells were then harvested and used for further analysis (Oil Red O staining, RT-qPCR, Western blot). Cells cultured with DMSO or without vehicle served as controls.

### 4.9. Statistics

All data were statistically evaluated for normality and homogeneity of variance by the Shapiro–Wilk test and Levene’s test, respectively. Statistically significant difference among groups was determined using one-way analysis of variance (ANOVA), Kruskal–Wallis test, Welch ANOVA followed by multiple comparison tests using the Student–Newman–Keuls (S-N-K) test for one-way ANOVA, Tukey Honestly Significant Difference (Tukey HSD) test for Welch ANOVA, and Dunn’s test for the Kruskal–Wallis test, accordingly. Probability values given were *p* < 0.05 (*), *p* < 0.01 (**), and *p* < 0.001 (***), respectively. Statistical analyses were conducted using SPSS 19.0 for windows (SPSS Inc., Chicago, IL). Details about the statistics are given in Appendix A.

## 5. Conclusions

LCN2 is a hepatoprotective factor upregulated during ER stress. It protects hepatocytes from being overwhelmed by UPR. The lack of LCN2 is associated with increased expression of CHOP in hepatocytes during ER stress response and increased apoptosis in acute toxic liver injury models. Consequently, ER stress, UPR, steatosis, and hepatocyte apoptosis after CCl_4_ application are slightly higher in *Lcn2*^−/−^ mice compared to wild type mice. It is accompanied by higher mRNA expression of ER stress-associated genes, higher quantities of mitochondrial proteins Bax and Cytochrome, elevated levels of cleaved Caspase-9, and slightly reduced amounts of Bcl2. These findings indicate that ER stress-mediated cell death in respective models is executed by the canonical mitochondrial apoptosis pathway. In primary hepatocyte cultures, the upregulation of LCN2 protects cellular integrity by maintaining homeostasis under ER stress conditions. In line, the lack of LCN2 results in higher susceptibility for hepatocyte damage during CCl_4_ intoxication. We therefore conclude that LCN2 is generated in hepatocytes as an acute phase protein under stress and harmful conditions and proved to protect hepatocytes in both acute and chronic ER stress induced apoptosis.

## Figures and Tables

**Figure 1 ijms-21-05230-f001:**
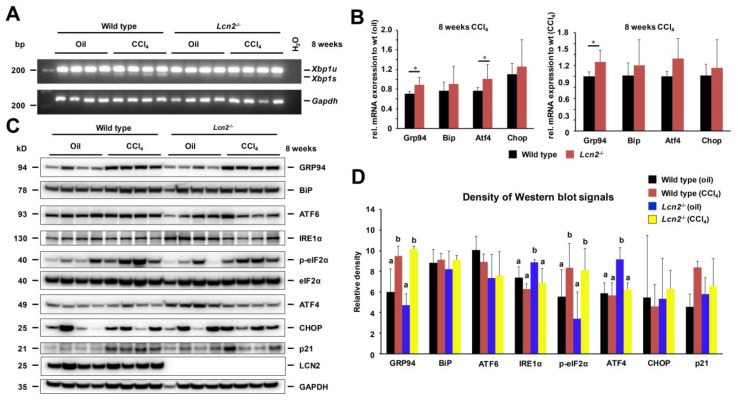
Long-term repeated CCl_4_ intoxication induced endoplasmic reticulum (ER) stress and unfolded protein responses (UPR). (**A**) Semi-quantitative PCR showed *Xbp1s* in CCl_4_-treated livers of wild type and *Lcn2*^−/−^ mice. (**B**) Real-time quantitative polymerase chain reaction (RT-qPCR) showing the ER stress-markers *Grp94*, *Bip*, *Atf4*, and *Chop* mRNA in relation to oil- (left panel) or CCl_4_-treated wild type mice (right panel). *Lcn2*^−/−^ mice showed relative higher values compared to the wild type. (**C**) Western blot analysis showing increased GRP94, BiP, p-eIF2α, and p21 in CCl_4_-treated mice, while showing lower levels of ATF6, IRE1α, and ATF4 in CCl_4_-treated *Lcn2*^−/−^ mice. CHOP levels fluctuate between animals. (**D**) Graph depicting Western blot quantifications. Details about the statistics are given in Appendix A.

**Figure 2 ijms-21-05230-f002:**
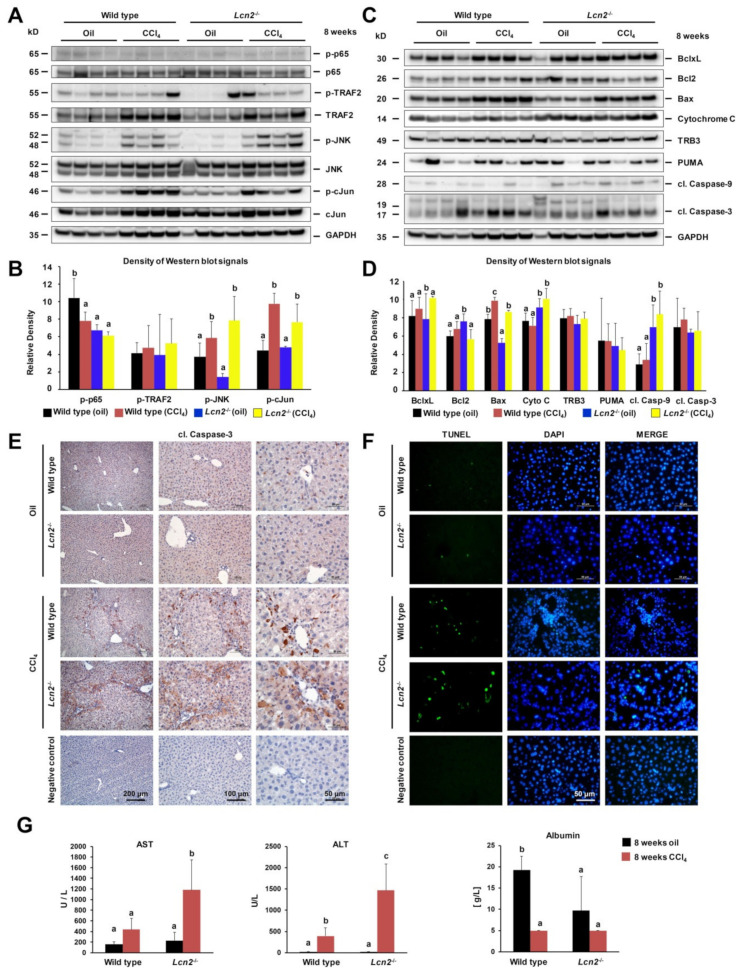
Chronic CCl_4_-induced hepatocyte apoptosis via JNK/c-Jun and mitochondrial pathways. (**A**) Western blot analysis and (**B**) quantification of p65 NF-κB, TRAF2, p-JNK, JNK, p-c-Jun, and c-Jun signaling cascades with GAPDH as a loading control, showing significant JNK/c-Jun activation in CCl_4_-treated mice. (**C**) Western blot analysis and (**D**) quantification showing Bcl2 family proteins Bcl-xL, Bcl2, Bax, and PUMA. Bax levels were higher in both types of CCl_4_-treated mice, with lower levels of Bcl2 in CCl_4_-treated *Lcn2*^−/−^ livers, with fluctuating levels of PUMA. Cleaved Caspase-9 levels were higher in oil- and CCl_4_-treated *Lcn2*^−/−^ mice, but TRB3 and cleaved Caspase-3 levels were increased in CCl_4_ administered livers, with GAPDH loading controls. (**E**) Immunohistochemistry staining of cleaved caspase-3 depicted positive signals distributed homogenously in non-parenchymal liver cells after mineral oil administration. In contrast, CCl_4_-treated wild type livers showed caspase-3 staining in infiltrating inflammatory macrophages around hepatic central veins, while in *Lcn2*^−/−^ mice cleaved caspase-3 was detected in both inflammatory cells and surrounding hepatocytes. (**F**) Terminal deoxynucleotidyl transferase dUTP nick end labeling (TUNEL)-stained paraffin slides showing TUNEL-positive infiltrating inflammatory cells around hepatic central veins of CCl_4_-treated wild type livers. *Lcn2*^−/−^ livers showing TUNEL-positive cells scattered in hepatocytes zone 3 with minimal TUNEL-positive cells after mineral oil administration. (**G**) Serum aspartate transaminase (AST) and alanine transaminase (ALT) levels were significantly higher in CCl_4_-treated *Lcn2*^−/−^ mice, while serum albumin levels showed decreases in CCl_4_-treated mice. Details about the statistics are given in Appendix A.

**Figure 3 ijms-21-05230-f003:**
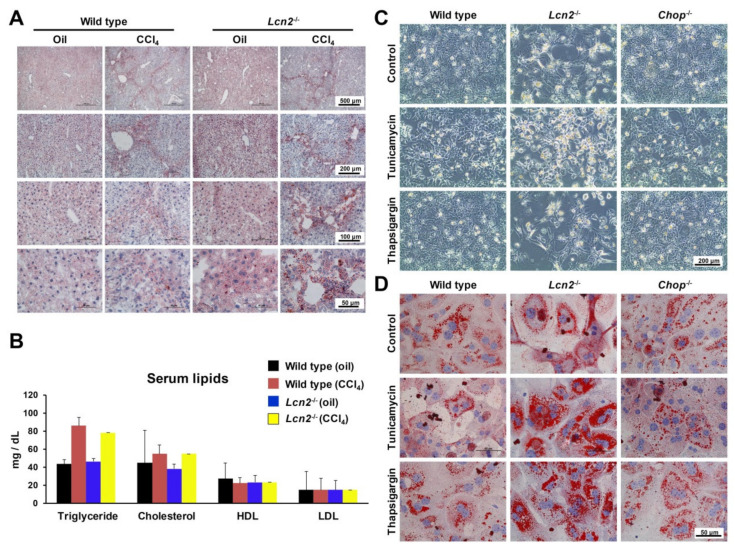
ER stress-induced steatosis. (**A**) Oil Red O-stained liver slides showing slightly increased amounts of lipid droplets in oil- and CCl_4_-injected *Lcn2*^−/−^ mice compared to wild type animals. (**B**) Serum triglyceride levels in CCl_4_-treated *Lcn2*^−/−^ mice had a tendency to be slightly lower than in wild type due to triglyceride accumulation and steatosis. No changes in serum high-density lipoprotein (HDL) and low-density lipoprotein (LDL) were observed. Details about the statistics are given in Appendix A. (**C**) Phase contrast microscopic pictures of cultured primary hepatocytes isolated from wild type, *Lcn2*^−/−^, and *Chop*^−/−^ mice treated with tunicamycin (TM) and thapsigargin (TG) for 36 h. *Lcn2*^−/−^ hepatocytes appeared to detach from the culture plates and be less confluent than wild type and *Chop*^−/−^. (**D**) Oil Red O-stained cultured primary hepatocytes treated with TM or TG for 36 h showing *Lcn2*^−/−^ hepatocytes to already contain more lipid droplets than the wild type and *Chop*^−/−^ hepatocytes, followed by 36 h TM or TG treatments showing significant enhanced lipid droplet formation.

**Figure 4 ijms-21-05230-f004:**
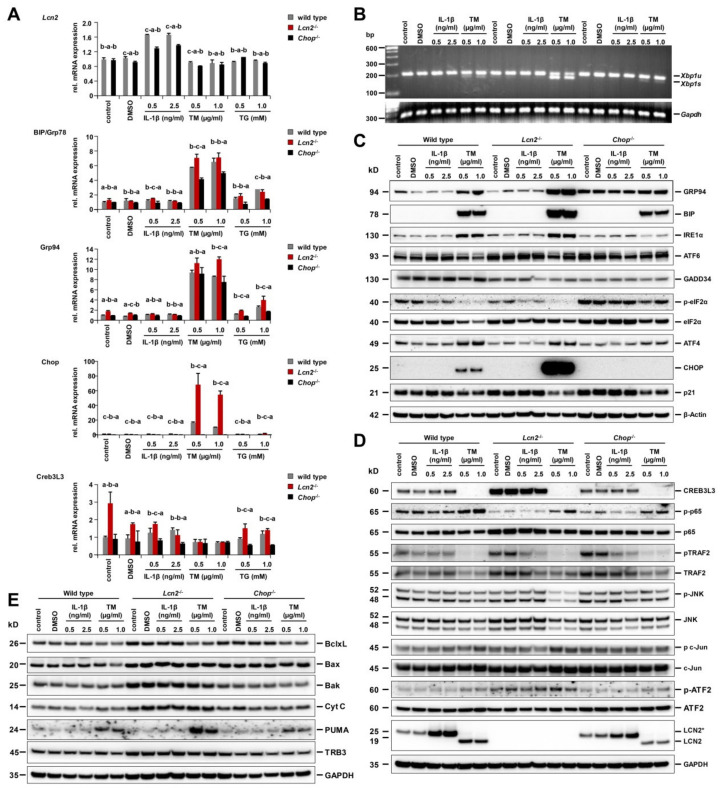
*Lcn2*^−/−^ hepatocytes reflecting strongest ER stress responses. (**A**) Real-time quantitative polymerase chain reaction (RT-qPCR) showing *Lcn2* mRNA and ER stress-markers *Bip*/*Grp78*, *Grp94*, *Chop*, and *Creb3L3* upon 24 h incubation with IL-1β, tunicamycin (TM), and thapsigargin (TG). IL-1β significantly induced *Lcn2* mRNA expression, while *Bip*/*Grp78*, *Grp94*, and *Chop* mRNA levels were also markedly increased by TM or TG stimulation, while *Lcn2*^−/−^ hepatocytes showed the highest levels. Additionally, the *Lcn2*^−/−^ hepatocytes possessed already high basal levels of *Creb3L3*. Details about the statistics are given in Appendix A. (**B**) Semi-quantitative PCR depicting unspliced (*Xbp1u*) and spliced *Xbp1* (*Xbp1s*) mRNA. *Lcn2*^−/−^ hepatocytes incubated with TM showing the most significant amounts of *Xbp1s*. (**C**) Western blot analysis of ER stress marker proteins GRP94, BiP, IRE1α, ATF6, GADD34, p-eIF2α, eIF2α, ATF4, and CHOP with β-actin as loading controls. *Lcn2*^−/−^ hepatocytes showed the highest ER stress responses with huge amounts of CHOP, while *Chop*^−/−^ hepatocytes increased higher baselines of p-eIF2α with decreased GADD34 and other UPR signaling proteins. (**D**) TM-induced ER stress was showed to activate and cleave CREB3L3/CREBH. TM activated NF-κB p65, TRAF2/JNK/c-Jun, and further inhibited LCN2 glycosylation, as evidenced by shifting of the LCN2 molecular weight, with GAPDH as loading controls. (**E**) TM altered expression of Bcl2 family proteins, *Lcn2*^−/−^ hepatocytes were clearly demonstrated to decrease Bcl-xL, and increase Bax, Bak, and PUMA with higher Cytochrome c levels. GAPDH was applied as loading control.

**Figure 5 ijms-21-05230-f005:**
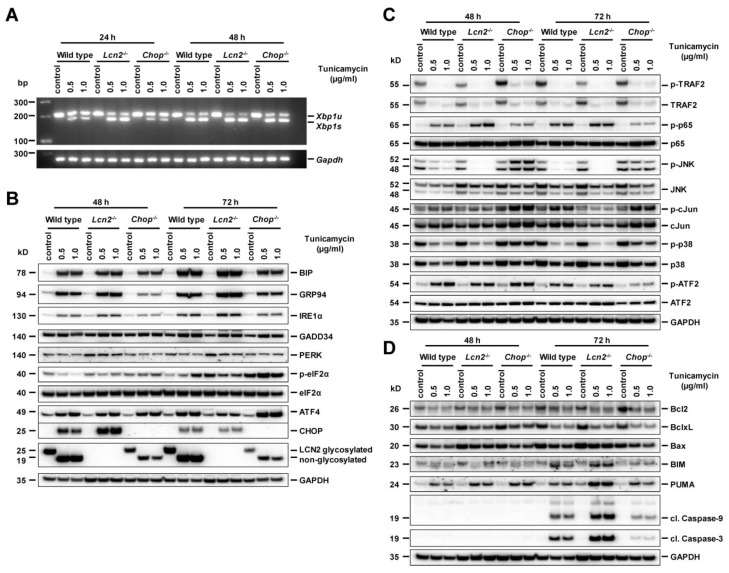
Tunicamycin incubation for 48 and 72 h did induce hepatocyte apoptosis. (**A**) Semi-quantitative PCR showing unspliced (*Xbp1u*) and spliced *Xbp1* (*Xbp1s*) mRNA after 24 and 48 h tunicamycin (TM) incubation. The levels of *Xbp1s* mRNA became almost level in all types of hepatocytes at 48 h. (**B**) Western blots of UPR signaling proteins BiP, GRP94, IRE1α, ATF6, PERK, p-eIF2α, eIF2α, ATF4, and CHOP. High amounts of CHOP in *Lcn2*^−/−^ declined comparable to wild type levels at 72 h. Accumulation of glycosylated and non-glycosylated LCN2 proteins in wild type showed significant uptick compared to *Chop*^−/−^ hepatocytes in 48 and 72 h TM incubation, with GAPDH as loading control. (**C**) Depicting TM induction of p65 NF-κB, TRAF2, JNK, and p38 MAPK activations. Phosphorylation of JNK/c-Jun and p38 was markedly increased in *Chop*^−/−^ hepatocytes, while the signals in wild type and *Lcn2*^−/−^ already declined. The ATF2 phosphorylation remained persistent in all types of hepatocytes, but were slightly lower in *Chop*^−/−^ at 72 h. (**D**) Bcl2 family protein expression showed decreased Bcl2 and Bcl-xL levels in all types of hepatocytes, while Bax was upregulated in *Lcn2*^−/−^. BH3-only proteins BIM and PUMA increased with highest levels in *Lcn2*^−/−^ hepatocytes well in correspondence to the levels of cleaved Caspase-9 and 3, but minimal in *Chop*^−/−^ hepatocytes. GAPDH served as loading control.

**Figure 6 ijms-21-05230-f006:**
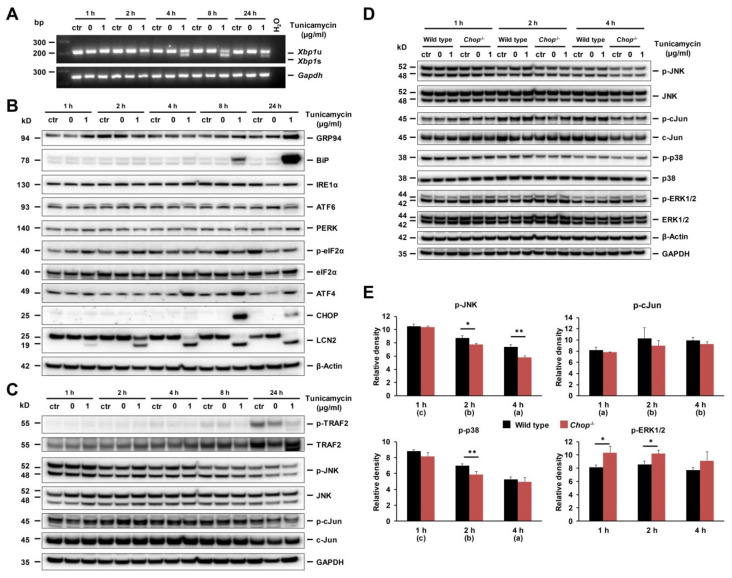
ER stress and UPR in the early stage of tunicamycin (TM) stimulated primary hepatocytes. (**A**) Semi-quantitative PCR of spliced *Xbp1* (*Xbp1s*) mRNA of wild type hepatocytes detected as early as 2 h upon TM incubation and increasing over time. (**B**) Western blot analysis of ER stress marker proteins GRP94, BiP, IRE1α, ATF6, PERK, p-eIF2α, ATF4, and CHOP, with blockade of LCN2 glycosylation. In this analysis, GAPDH served as loading controls. (**C**) Upon 1 h TM stimulation, phosphorylated JNK reduced continuously, including p-c-Jun, but at a slower pace corresponding to p-JNK, with GAPDH as loading control. (**D**) Baseline levels of p-JNK, p-c-Jun and p-38 in *Chop*^−/−^ hepatocytes showed lower compared to wild type and decreasing over time, while p-ERK1/2 remained high. (**E**) Western blot density graph from (**D**)**.** Details about the statistics are given in Appendix A.

**Table 1 ijms-21-05230-t001:** Primers used in this study.

Murine Gene	Acc. No.	Primer
*Xbp-1*	NM_001271730	for: 5′-GAACCAGGAGTTAAGAACACG-3′
		rev: 5′-AGGCAACAGTGTCAGAGTCC-3′
		*Xbp1u* = 205 bp, *Xbp1s* = 179 bp
*Gapdh*	NM_008084	for: 5′-TCGTGGATCTGACGTGCCGCCTG-3′
		rev: 5′-CACCACCCTGTTGCTGTAGCCGTAT-3′
		semi-qPCR = 251 bp
*Bip*	NM_001163434	for: 5′-CTGAGGCGTATTTGGGAAAG-3′
		rev: 5′-TCATGACATTCAGTCCAGCAA-3′
*Grp94*	NM_011631	for: 5′-AGGGTCCTGTGGGTGTTG-3′
		rev: 5′-CATCATCAGCTCTGACGAACC-3′
*Chop*	NM_007837	for: 5′-GCGACAGAGCCAGAATAACA-3′
		rev: 5′-GATGCACTTCCTTCTGGAACA-3′
*Creb3L3*	NM_145365	for: 5′-CTCCCGCTTCAACCTCACT-3′
		rev: 5′-GCCAAGGAATGCTGTTGC-3′
*Atf4*	NM_001287180	for: 5′-GGCTGGTCGTCAACCTATAAA-3′
		rev: 5′-CAGGCACTGCTGCCTCTAAT-3′

**Table 2 ijms-21-05230-t002:** Antibodies used in this study.

Antibody	Cat. No.	Supplier	Dilution
ATF2	242	Santa Cruz, Santa Cruz Biotech, CA, USA	1:1000
p-ATF2	8398	Santa Cruz	1:1000
ATF4	sc-390063	Santa Cruz	1:1000
ATF6	sc-166659	Santa Cruz	1:1000
β-actin	A5441	Sigma, Taufkirchen, Germany	1:10,000
Bak	12105	Cell Signaling, Darmstadt, Germany	1:1000
Bax	14796	Cell Signaling	1:1000
Bcl-xL	2764	Cell Signaling	1:1000
Bcl2	3498	Cell Signaling	1:1000
BIM	374358	Santa Cruz	1:1000
BiP	3177	Cell Signaling	1:1000
c-Jun	9165	Cell Signaling	1:1000
CHOP/GADD153	sc-7351	Santa Cruz	1:1000
cleaved Caspase-3	9664	Cell Signaling	1:1000
cleaved Caspase-9	9507	Cell Signaling	1:1000
CREB3L3/CREBH	sc-377331	Santa Cruz	1:1000
Cytochrome C	11940	Cell Signaling	1:1000
eIF2α	9722	Cell Signaling	1:1000
p-eIF2α	3597	Cell Signaling	1:1000
ERK1/2	9102	Cell Signaling	1:1000
p-ERK1/2	9101	Cell Signaling	1:1000
GADD34	373815	Santa Cruz	1:1000
GAPDH	sc-32233	Santa Cruz	1:1000
GRP94	sc-393402	Santa Cruz	1:1000
IRE1α	3294	Cell Signaling	1:1000
LCN2	AF1857	R&D Systems, Wiesbaden, Germany	1:1000
p21	556430	BD Bioscience, Heidelberg, Germany	1:1000
p38	612168	BD Biosciences	1:2500
p-p38	612288	BD Biosciences	1:2500
p65	sc-8008	Santa Cruz	1:1000
p-p65	3033	Cell Signaling	1:1000
p-c-Jun	3270	Cell Signaling	1:1000
PERK	377400	Santa Cruz	1:1000
PUMA	374223	Santa Cruz	1:1000
SAPK/JNK2	9258	Cell Signaling	1:1000
p-SAPK/JNK	4668	Cell Signaling	1:1000
TRAF2	4712	Cell Signaling	1:1000
p-TRAF2	13908	Cell Signaling	1:1000
TRB3	390242	Santa Cruz	1:1000

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
