# Peer review of "Chronic Carbon Tetrachloride Applications Induced Hepatocyte Apoptosis in Lipocalin 2 Null Mice through Endoplasmic Reticulum Stress and Unfolded Protein Response"

_ijms, 2020, doi:10.3390/ijms21155230_

Round 1
Reviewer 1 Report
The article submitted for evaluation addresses an interesting topic. However, it needs major changes before it is considered for publication. The work would gain in being more focused, showing the main results that support the authors' hypothesis. Given the amount of results obtained in experiments carried out under different conditions, with different models, the diversity of parameters analyzed, the great variability of results, it is difficult to follow and understand the main message. Authors should also review the text in order to make it clearer and identify the novelty of the study and what are the main conclusions. Statistical analysis also requires revision since the t test is used for multiple comparisons, which is not correct.
Author Response
Dear reviewer,
Many thanks for the time you spent in reading our manuscript. We are grateful for your
valuable comments. In the revised version, we have tried to improve the presentation. In
particular, we have added a Conclusion section in which we added some brief sentences
summarizing the main messages of our study.
In regard to the statistics we fully agree. As suggested and after consultations with an expert
statistician, we have now revised our statistics. As you mentioned correctly, comparing more
than two groups (in Figures 1D, 2B, 2D, 2G, 3B, 4A, and 6E) does not allow using t-test. We
have now used different tests for multiple comparisons and added new Annex A, providing
comprehensive details of our statistical analysis. In addition, we have modified the respective
part in the Materials and Methods section. Nevertheless, the previous significances have not
changed by using these tests. Please note: Based on her helpful input, we have added the
statistician to the author list (Professor Anothai Trevanich, Department of Statistics, Khon
Kaen University). A necessary filled-in form “Change of authorship” was uploaded to the
journal’s homepage with our submission.
We hope you will agree that our work is now suitable for publication in the International
Journal of Molecular Science.
Best regards
Erawan Borkham-Kamphorst and Ralf Weiskirchen
Reviewer 2 Report
The authors need to justify that how LCN2 fits into the whole picture of ER stress signaling pathway and how LCN2 differentially regulates ER stress under acute and chronic inflammatory conditions. In other words, explanation of the molecular mechanism of LCN2-involved ER stress signaling is needed into the introduction.
In figure 1, ER stress makers including GRP78 (Bip), ATF6, CHOP were not up-regulated under chronic CCl4 application, which was contrary to the statement made in the paper about enhanced ER stress and unfolded protein responses (UPR) in the same condition. Although other molecules including GRP94, ATF4, and p-eIF2α showed statistical significance between wild type and CCl4 application, data itself seemly did not support the conclusion.
In figure 4C, TM-induced ER stress response in LCN2 null mice hepatocytes provided better results compared to wild control and that in observed animal data. Any comments on why data from in vitro experiments did give better results?
In figure 6E, phosphorylation of protein analysis such as p-JNK, p-c-Jun, p-ERK1/2 did not show significance at indicated time points. Generally speaking, phosphorylation of protein is supposed to be a transient and short time period, which usually lasts for minutes. Earlier time point may be considered, such as 5 minutes, 10 minutes, et al.
If LCN2 protects hepatocytes in both acute and chronic ER stress-induced apoptosis, how to explain its increase in ASH and NASH to promote liver injury?
Minor points:
1. GADD153 in figure 5B is equivalent to CHOP, please be consistent.
2. Figure 3 figure legend, Oil red O staining is miswritten as C.
Author Response
The authors need to justify that how LCN2 fits into the whole picture of ER stress signaling
pathway and how LCN2 differentially regulates ER stress under acute and chronic
inflammatory conditions. In other words, explanation of the molecular mechanism of LCN2-
involved ER stress signaling is needed into the introduction.
Dear reviewer,
First of all we would greatly acknowledge your help in improving our manuscript. We fully
agree that an explanation about the molecular mechanisms of LCN2-involved ER-signaling
was missing in the previous version of our manuscript. Following your advice, we have
added two sentences in the revised version clarifying this issue.
In figure 1, ER stress makers including GRP78 (Bip), ATF6, CHOP were not up-regulated
under chronic CCl4 application, which was contrary to the statement made in the paper
about enhanced ER stress and unfolded protein responses (UPR) in the same condition.
Although other molecules including GRP94, ATF4, and p-eIF2α showed statistical
significance between wild type and CCl4 application, data itself seemly did not support the
conclusion.
Many thanks for this comment. In our study, we have analyzed the expression of respective
genes by RT-qPCR (Figure 1B) and Western blot analysis (Figures 1C and 1D). We agree
with your concern that we had not clearly stated if the expression is altered at mRNA or
protein level. In the revised version, we have made some adjustment in the respective text
passage and added a short comment at the beginning of Discussion section highlighting this
issue. Please note that based on the comments of reviewer 1, it was also necessary to revise
the statistics in our paper. He/she mentioned that the usage of t-test in Figure 1D is not
meaningful because we compared more than two groups. As he/she suggested, we have
done other statistical tests for which details are presented in new Annex A.
In figure 4C, TM-induced ER stress response in LCN2 null mice hepatocytes provided better
results compared to wild control and that in observed animal data. Any comments on why
data from in vitro experiments did give better results?
You mentioned a valid point. TM-induced ER stress responses in hepatocytes are better in
Lcn2 null mice than in cells isolated from control mice. In addition, our in vitro data showed
better results than our in vivo experiments. In our view there are some reasons that can
explain these findings. We have previously shown that LCN2 acts as a protective factor in
liver homeostasis (Borkham-Kamphorst et al., Biochim. Biophys. Acta 2013;1832:660-73).
This means that cells lacking LCN2 are more prone to injury, explaining why these cells
respond more to additional triggers such as TM. In regard to the comparison of in vitro and in
vivo data we think that in vivo there are many mechanisms preventing and/or counteracting
overshooting experimental hepatic damage.
To address your concern we have added some short comments at the end of the Discussion
section.
In figure 6E, phosphorylation of protein analysis such as p-JNK, p-c-Jun, p-ERK1/2 did not
show significance at indicated time points. Generally speaking, phosphorylation of protein is
supposed to be a transient and short time period, which usually lasts for minutes. Earlier time
point may be considered, such as 5 minutes, 10 minutes, et al.
In Figure 6E we showed that phosphorylation of JNK, p38 and pERK is significantly
increased after 2 hours of TM treatment. We agree that protein phosphorylation is usually
analyzed at earlier time points when stimulating cells with agents (cytokines, chemokines)
have specific receptors. However, TM is a mixture of homologous nucleoside antibiotics that
inhibits protein glycosylation. To inhibit N-linked glycosylation it is necessary to incubate the
cells longer to induce the ER stress machinery. In a previous study (Borkham-Kamphorst et
al., Cell. Signal 2019;55:90-9), we have established the concentrations and time points that
are most suitable to induce ER stress in cultured hepatocytes.
To address your comment, we have added a short comment in which we clarify why we used
the respective concentrations and time points for our TM experiments. We agree that these
comments are helpful for the reader.
If LCN2 protects hepatocytes in both acute and chronic ER stress-induced apoptosis, how to
explain its increase in ASH and NASH to promote liver injury?
We agree that this is a key question. Indeed, previous work from others has shown that
LCN2 exacerbates steatohepatitis in an animal model of ASH/NASH by promoting
neutrophil-macrophage crosstalk (Ye et al., J. Hepatol. 2016;65:988-97). However, in
contrary to this findings, LCN2 protects against diet-induced nonalcoholic fatty liver disease
by targeting hepatocytes (Xu et al., Hepatol. Commun. 2019;3:763-75) or by interfering with
expression of the lipid droplet associate protein PLIN5 (Asimakopoulou et al., Biochim
Biophys Acta 2014;1842:1513-24). Moreover, there are also finding showing that LCN2 plays
a protective role against the progression of hepatocellular carcinoma by suppressing cell
proliferation and invasion (Lee et al., Int. J. Oncol., 2011;38:325-33).This shows that the
functionality of LCN2 depends on the condition analyzed. Nevertheless, it should be kept in
mind that LCN2 was originally isolated as an acute phase protein (Liu and Nilsen-Hamilton,
J. Biol. Chem. 1995;270:22565-70) that are important mediators produced by the liver during
acute and chronic inflammatory states. These factors are regarded as protective factors
having functions in opsonization of micro-organisms and their products, activating
complement, binding cellular remnants, neutralizing enzymes, scavenging free radicals, and
in modulating the host’s immune response. Therefore, we think that LCN2 have more
protective than harmful influences on liver health. We have added some short comment on
this issue to address your comment at the end of the Discussion section.
Minor points:
1. GADD153 in figure 5B is equivalent to CHOP, please be consistent.
Many thanks for this comment. As suggested, we have modified GADD153 to CHOP in
respective figure.
2. Figure 3 figure legend, Oil red O staining is miswritten as C.
Many thanks for your special attention in reading. We have changed the respective lettering
from (C) to (D).
We hope you will agree that our work is now suitable for publication in the International
Journal of Molecular Science.
Best regards
Erawan Borkham-Kamphorst and Ralf Weiskirchen
Round 2
Reviewer 1 Report
The authors have made substantial alterations to the manuscript that made it more clear, including conclusions sections. The revision of the statistical analysis sections improved the quality and the confidence in the study.
Reviewer 2 Report
None.